# A noncanonical chaperone interacts with drug efflux pumps during their assembly into bacterial outer membranes

Christopher J. Stubenrauch[1,2]*, Rebecca S. Bamert[1,2], Jiawei Wang[1,2¤], Trevor Lithgow[1,2]*

1 Infection and Immunity Program, Biomedicine Discovery Institute and Department of Microbiology, Monash University, Clayton, Victoria, Australia, 2 Centre to Impact AMR, Monash University, Clayton, Victoria, Australia

¤ Current address: European Molecular Biology Laboratory, European Bioinformatics Institute (EMBL-EBI), Wellcome Genome Campus, Hinxton, Cambridge, United Kingdom

* christopher.stubenrauch@monash.edu (CJS); trevor.lithgow@monash.edu (TL)

## Abstract

Bacteria have membrane-spanning efflux pumps to secrete toxic compounds ranging from heavy metal ions to organic chemicals, including antibiotic drugs. The overall architecture of these efflux pumps is highly conserved: with an inner membrane energy-transducing subunit coupled via an adaptor protein to an outer membrane conduit subunit that enables toxic compounds to be expelled into the environment. Here, we map the distribution of efflux pumps across bacterial lineages to show these proteins are more widespread than previously recognised. Complex phylogenetics support the concept that gene cassettes encoding the subunits for these pumps are commonly acquired by horizontal gene transfer. Using TolC as a model protein, we demonstrate that assembly of conduit subunits into the outer membrane uses the chaperone TAM to physically organise the membrane-embedded staves of the conduit subunit of the efflux pump. The characteristics of this assembly pathway have impact for the acquisition of efflux pumps across bacterial species and for the development of new antimicrobial compounds that inhibit efflux pump function.

## Introduction

One of the most commonly deployed mechanisms of antimicrobial resistance (AMR) is mediated by bacterial efflux pumps, protein complexes that have evolved to rid cells of toxic compounds [1]. The fundamental mechanism by which these efflux pumps function in gram-negative bacteria depends on a tripartite architecture, comprised of 3 components that combine to span the inner membrane, periplasm, and outer membrane (Fig 1A), as understood from several landmark structural studies on these tripartite complexes [2–5]. Most of the structural and functional work has been focussed on the prototypical pumps AcrAB-TolC and MexAB-OprM. These pumps are constitutively expressed in *Escherichia coli* and *Pseudomonas aeruginosa*, respectively, and are the workhorses that confer low levels of resistance to clinically relevant drugs before more specialised resistance mechanisms can evolve [6].

**Data Availability Statement:** All relevant data are within the paper and its Supporting Information files.

**Funding:** This work was supported by a National Health and Medical Research Council (https://www.nhmrc.gov.au/) Program Grant (1092262 to T.L.). The funders had no role in study design, data collection and analysis, decision to publish, or preparation of the manuscript.

**Competing interests:** The authors have declared that no competing interests exist.

**Abbreviations:** AMR, antimicrobial resistance; BAM, β-barrel assembly machinery; HGT, horizontal gene transfer; LPS, lipopolysaccharide; MARP, Monash Animal Research Platform; OEP, outer membrane efflux protein; TAM, translocation and assembly machinery.

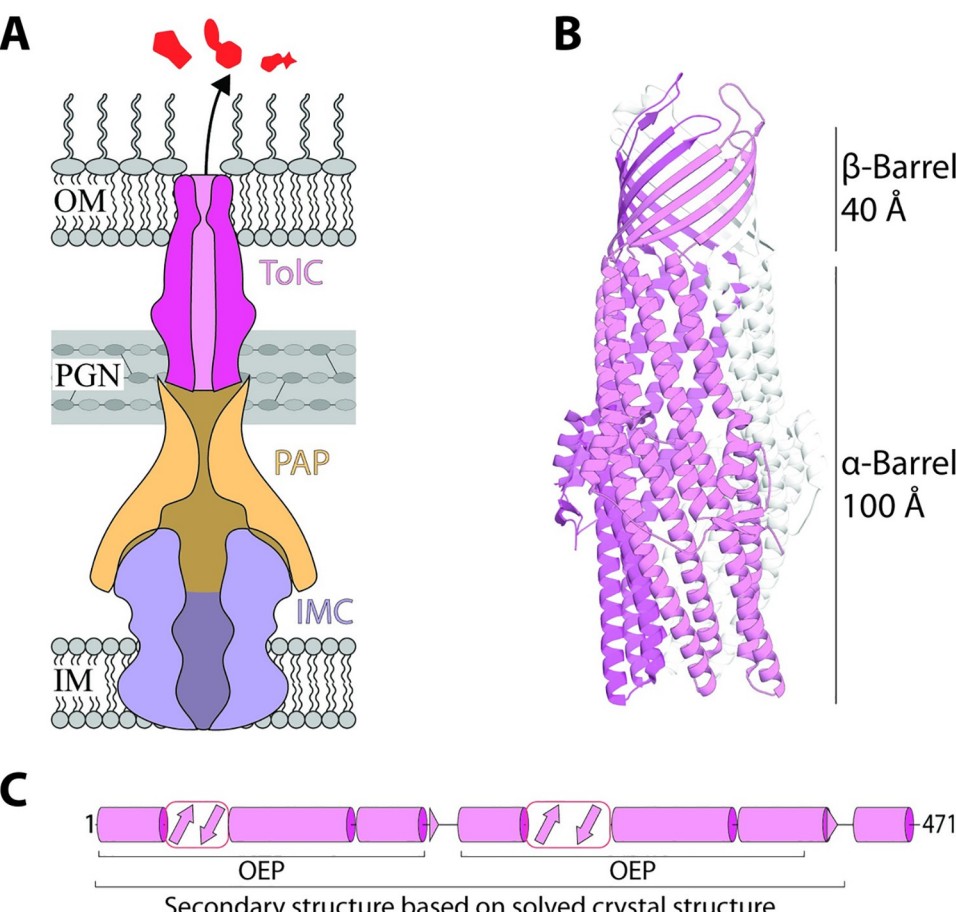

**Fig 1. Tripartite efflux pumps.** (A) Schematic of an efflux pump removing harmful substances. The pump comprises an IMC protein from the RND, ABC, or MFS superfamilies, a PAP, and an OMF family component, like TolC [1]. (B) Ribbon diagram of TolC (PDB: 1EK9). Sizes of each structural domain are indicated to the right [7]. One of the 3 monomers is coloured pink for clarity. (C) Structural map of one TolC monomer (without its signal peptide), based on its crystal structure (PDB: 1EK9) over residues 1–428 and as predicted using PSIPRED 4.0 [8] over residues 429–471. The characteristic OEP regions are indicated. Cylinders represent α-helices; arrows represent β-strands. IMC, inner membrane channel; OEP, outer membrane efflux protein; OMF, outer membrane factor; PAP, periplasmic adaptor protein; PGN, peptidoglycan.

TolC is the archetypal conduit of tripartite efflux pumps and contains a characteristic αβ-barrel architecture (Fig 1B). To achieve this structure, the TolC polypeptide needs to be folded into staves, 3 of which are assembled together to form a transmembrane β-barrel embedded in the outer membrane (Fig 1C) as well as an α-barrel that extends 100 Å into the periplasm [7]. In *E. coli*, this trimeric TolC conduit subunit is then engaged either onto AcrAB or onto one of 8 other distinct efflux pumps [9]. In other bacterial species, multiple proteins with structural homology to TolC exist, such as in *Pseudomonas* species, where at least 18 of these TolC-like proteins have been identified as purpose-evolved conduits [10]. Through horizontal gene transfer (HGT), the genes encoding TolC-like proteins, and their cognate partner proteins, can be acquired across diverse bacterial lineages promoting the spread of AMR phenotypes [11–15]. The acquisition of these genes is the first committed step to having a functional pump, but the phenotype can only be enacted if the complex folding pathway for the conduit subunit can be mediated in the host species acquiring those genes. Catalysing the assembly pathway for TolC-like proteins would depend on the host having the necessary chaperones to

drive the protein folding and assembly and might be limited by sequence diversity in the acquired polypeptide, which did not coevolve with those chaperones [16].

The major, membrane-embedded chaperone catalysing outer membrane protein assembly is the β-barrel assembly machinery (BAM) complex, which works together with the translocation and assembly machinery (TAM) [17]. Architecturally, the BAM complex is composed from a protein of the Omp85 superfamily, BamA, and a set of accessory proteins the identity of which vary across the lineages of gram-negative bacteria [18–24]. The TAM is a second module of the BAM composed of an inner membrane protein, TamB, and a protein of the Omp85 superfamily, TamA, in the outer membrane [20,25–27]. The TAM has a more limited phylogenetic distribution than the BAM complex [28] and has only been characterised functionally in 3 bacterial species: *E. coli*, *Citrobacter rodentium*, and *Klebsiella pneumoniae* [25–27,29].

TolC-like proteins are topologically unlike any other substrate handled by the BAM complex or the TAM, because their β-barrel fold is not evolutionarily related to other outer membrane proteins, but instead evolved from the periplasmic component of the efflux pump, with convergent evolution bringing it to resemble the β-barrel fold [30]. As such, the assembly pathway for TolC-like proteins is far from simple. The membrane-embedded β-barrel of TolC is composed of 3 identical stave domains. Each stave is constructed through assembling an antiparallel array of 4 beta-strands into a TolC monomer that alone is not ideally suited to either the aqueous environment of the periplasm, nor the hydrophobic environment of the outer membrane. Each of these stave domains then needs to be brought together to construct the final β-barrel domain of the trimeric TolC protein (Fig 1B and 1C). While the BAM complex impacts on TolC assembly in *E. coli* [31,32], the BAM complex is paradoxically not involved in assembly of the TolC-like proteins OprM in *P. aeruginosa* [33] nor HgdD in the cyanobacterium *Anabaena* sp. PCC 7120 [34]. Mutant forms of the BAM complex (BamAΔR64 and BamAΔR36-K89), which impact the assembly of 14 well-characterised β-barrel proteins, do not affect TolC assembly [35,36], yet deletion of the BAM complex accessory lipoprotein BamB significantly increases TolC levels in the outer membrane [37] and also confers an increased TolC trimerisation rate [38]. Furthermore, the assembly kinetics of TolC trimerisation are, at least partially, reduced in the absence of the TAM chaperone, suggesting it to be important for TolC assembly but with no clear indication of the mechanism underlying the assembly process [38]. Together, these phenotypic data highlight the unusual nature of TolC assembly.

Here, we provide direct evidence of the interaction between TamA and TolC, where each β-strand of the TolC stave domain was found to interact with the lateral gate of TamA. Building on previous observations of mutant phenotypes [35,36,38], and incorporating new findings from in situ cross-linking and pulse-chase assays, we suggest a promiscuous assembly process for TolC-like proteins such that they can use either the BAM complex or the TAM for assembly. We discuss how this promiscuity would be highly relevant in enhancing the success rates with which HGT could establish functional efflux pumps in diverse lineages of bacteria. This study underscores the bottlenecks in acquisition, expression, and activity of drug efflux pumps and informs on the interplay between the BamA and TamA chaperones mediating outer membrane protein assembly.

## Results

### Establishing an assay using the lateral gate of TamA

TamA contains a lateral gate: an opening of the first and last transmembrane β-strands proposed to form a pathway for substrate insertion into the outer membrane [39,40]. To

determine whether the presumptive lateral gate of TamA does engage substrate polypeptides like TolC in vivo, we generated 2 TamA constructs that could be closed and locked to prevent substrate engagement (TamA-G250C/E555C and TamA-G252C/G553C) by introducing a cysteine substitution in the first (G250C or G252C) and last (G553C or E555C) transmembrane β-strands of TamA (Fig 2A). Under normal conditions, these TamA molecules coexist in both oxidation states, "locked" (oxidised) and "unlocked" (reduced), and are readily distinguished by nonreducing SDS-PAGE analysis because the disulphide bond will force "locked" TamA molecules into a compact conformation that migrates faster under electrophoresis (Fig 2B and 2C; S1 Fig). On average, we observed that 2/3 of TamA species were "locked" and would prevent the lateral gate from opening, whereas the remaining 1/3 of TamA molecules sample an "unlocked" conformation that would remain functional (Fig 2B and 2C; S1 Fig).

To assess whether these "locked" TamA species were nonfunctional, we performed a radiolabelling pulse chase assay that uses FimD assembly as a readout of TAM function [38,39,41] (S2A Fig). Previously, we have shown that the outer membrane β-barrel FimD rapidly assembles in the presence of the TAM and can be cleaved into 2 fragments using an extracellular protease, proteinase K [38] (first and third panels, S2B Fig). These fragments were identified by mass spectrometry [38] and correspond to an approximately 50-kDa N-terminal fragment (i.e., fragment A) and an approximately 40-kDa C-terminal fragment (i.e., fragment C), with the cleavage site located within the cell surface–exposed loop 7 of FimD (between residues 505 and 506: ETQ/DGV) [38]. In the absence of the TAM, FimD assembly is mediated solely by the BAM complex and proceeds via an assembly intermediate that is instead cleaved by extracellular proteinase K into an approximately 45-kDa fragment (i.e., fragment B) (second panel, S2B Fig). This fragment does not correspond to the N- nor C-terminal region of FimD but instead corresponds to a central region of FimD [38]. While native FimD eventually forms in the absence of the TAM, this occurs much more slowly, so the presence of the approximately 45-kDa fragment B is indicative of a nonfunctional TAM [38,39,41].

We first determined that single cysteine substitutions at the first or last transmembrane β-strand did not affect TamA function (first 2 panels, Fig 2D and 2E; first 4 lanes, S2C Fig). We then assessed whether a "locked" lateral gate would result in the generation of the approximately 45-kDa fragment characteristic of a nonfunctional TamA. Indeed, FimD was predominantly cleaved into the approximately 45-kDa fragment with small amounts of the approximately 40- and approximately 50-kDa fragments observed due to the presence of the 1/3 "unlocked" and functional TamA molecules (third panel, Fig 2D and 2E; lanes 5 to 6, S2C Fig). On addition of a reducing agent (5 mM DTT), which breaks the disulphide bond that would otherwise lock the lateral gate, only the fragments characteristic of correctly assembled FimD were observed (fourth panel, Fig 2D and 2E; lanes 7 to 8, S2C Fig). Conversely, on addition of an oxidising agent (100 μM CuSO$_4$), which promotes formation of the disulphide bond, a more substantial proportion of FimD was cleaved into the approximately 45-kDa fragment B, especially for the cells containing TamA-G252C-G553C (lanes 9 to 10, S2C Fig). Together, these data confirm that TamA contains a lateral gate that is required for function in vivo.

## TolC interacts with the lateral gate of TamA

We next wanted to assess whether TolC could interact with the lateral gate of TamA. To this end, we generated 4 constructs, each with a single cysteine substitution within one of its 4 transmembrane β-strands: TolC-G45C, TolC-S68C, TolC-S253C, and TolC-S287C (Fig 3A and 3B). Using erythromycin resistance as a measure of TolC function [42], we show that each TolC mutant can fully complement the Δ*tolC* phenotype to assemble the AcrAB-TolC pump

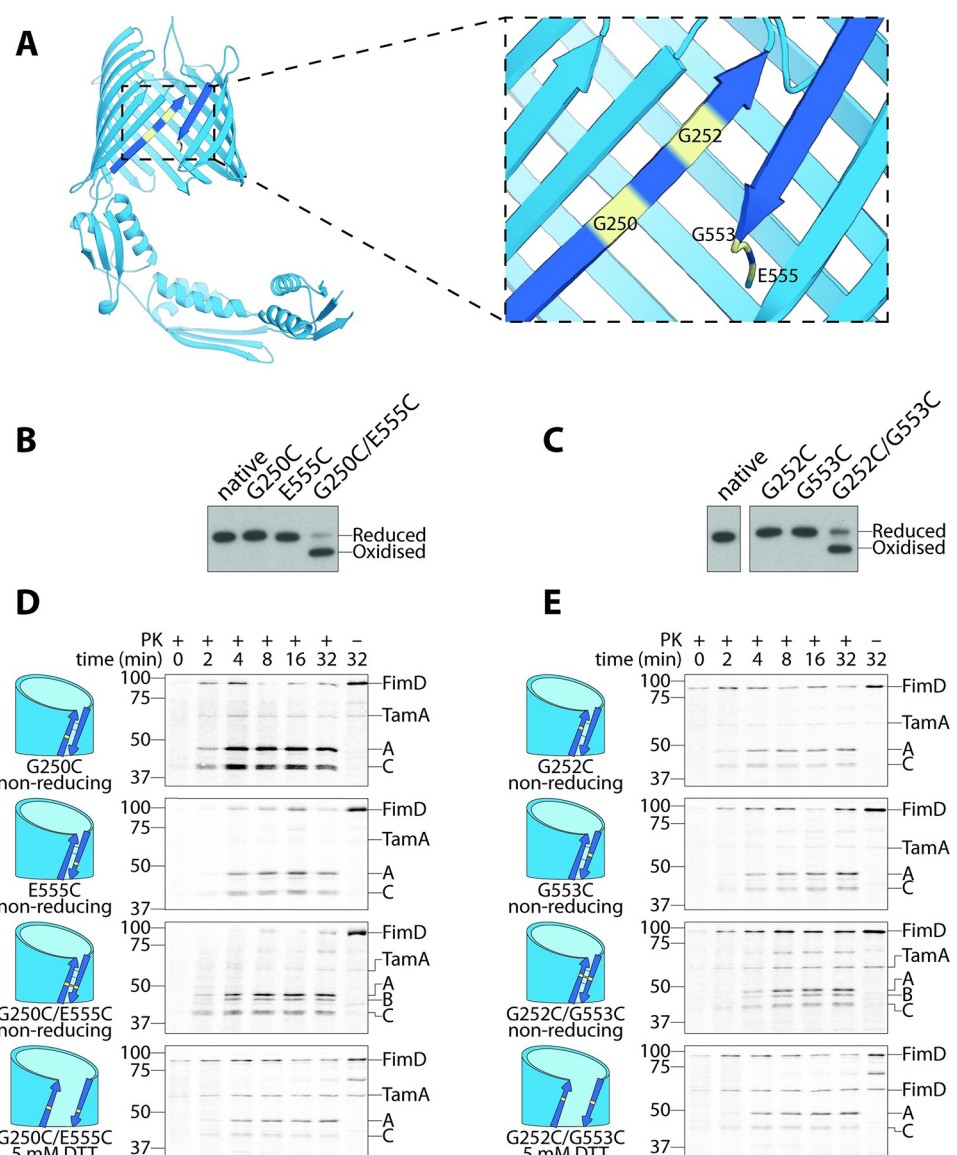

**Fig 2. TamA requires a lateral gate for function.** (A) Ribbon diagram of TamA (PDB: 4C00), where the first and last β-strands representing the lateral gate have been coloured a darker shade of blue. Each cysteine substitution is highlighted in yellow. (B-C) Cells prepared for pulse chase analysis were subjected to nonreducing SDS-PAGE and immunoblotting using antibodies raised against TamA. TamA oxidation states are indicated, based on their migration speeds (NB: Oxidised TamA is more compact because it cannot be fully denatured without a reducing agent and therefore migrates faster). (D-E) FimD assembly was monitored over time by pulse chase analysis in Δ*tamA* cells complemented with the indicated TamA cysteine mutants. Aliquots were taken at 10 s (0 min), 2, 4, 8, 16, and 32 min and treated with proteinase K (±PK). Total protein was analysed by SDS-PAGE and storage phosphor imaging. The position of FimD and its fragments A, B, and C are indicated on the right of autoradiograms, and the protein standards are indicated on the left (sizes are kDa). A (approximately 50 kDa) and C (approximately 40 kDa) represent the N- and C-terminal fragments of correctly assembled FimD, respectively, whereas B (approximately 45 kDa) is a central fragment of an assembly intermediate of FimD that accumulates in the absence of functional TamA [38]. Where reducing conditions were required, 5 mM DTT was used to supplement media during pulse chase analysis. Each pulse chase analysis was performed with at least 3 biological replicates. (B-E) Uncropped images are presented in S1 Raw Images; original autoradiographs and immunoblots (including relevant replicates) are presented in S2 and S3 Raw Images, respectively.

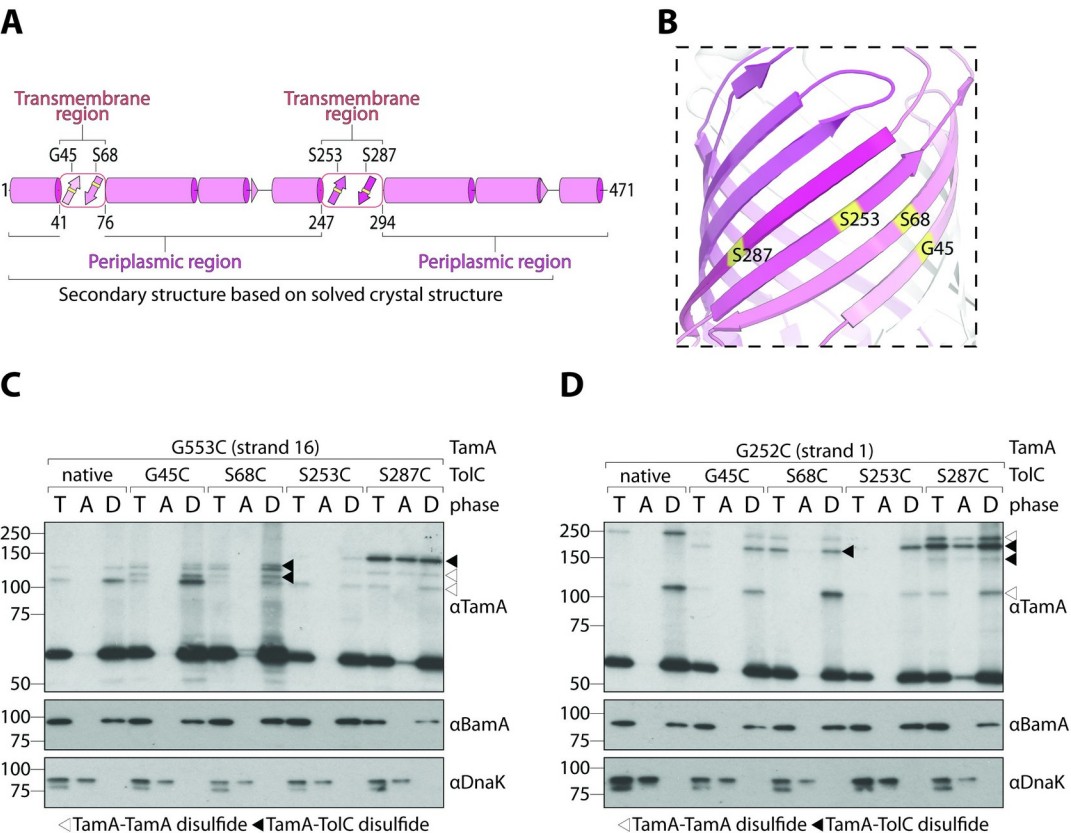

**Fig 3. TolC is a substrate of TamA.** (A) Structural map as in [Fig 1A](). The boundaries of the transmembrane domain are indicated below the structural map, and the positions of each cysteine substitution are indicated above. (B) Ribbon diagram of the TolC β-barrel domain (PDB: 1EK9). Each cysteine substitution is highlighted in yellow. Transmembrane β-strands are coloured as in panel (A). (C-D) Membranes were isolated from Δ*tamA* cells expressing the indicated *tolC* and *tamA* variants. Peripheral proteins (DnaK) were separated from integral proteins (BamA) by Triton X-114 extraction. Total membranes (T), aqueous phases (A), and detergent-enriched phases (D) were analysed by immunoblotting after nonreducing SDS-PAGE (TamA) or reducing SDS-PAGE (BamA and DnaK). White triangles indicate TamA–TamA disulphide interactions and black triangles indicate TamA–TolC interactions. This experiment was performed in biological triplicate (from 3 independent membrane preparations). Uncropped images are presented in S1 Raw Images; original immunoblots (including relevant replicates) are presented in S3 Raw Images.

and expel erythromycin at the level of the wild-type strain ([S1 Table]()). After confirming that each cysteine variant did not affect TolC function, we paired them with TamA-G553C (i.e., cysteine substitution in the last transmembrane β-strand). These pairings were made instead of using TamA-E555C because TamA-G252C/G553C was slightly more oxidised under nonreducing conditions ([S1 Fig]()), which may translate to stronger TamA–TolC disulphide interactions.

Membranes purified from *E. coli* expressing each construct pair were analysed by SDS-PAGE to assess whether higher molecular weight bands consistent with a TamA–TolC disulphide interaction could be observed ([Fig 3C]()). These membranes were devoid of periplasmic and cytoplasmic protein contaminants ([S3 Fig]()), and each contained higher molecular weight bands as assessed by nonreducing SDS-PAGE and immunoblotting with antibodies raised against TamA (black triangles, [Fig 3C]()). Unfortunately, when we assessed the band positions using antibodies raised against TolC ([S4 Fig]()), some of the TamA-TolC bands were obfuscated by what appears to be cysteine-mediated TolC-TolC bands, so we instead included a control set of membranes that contains TamA-G553C and native TolC to compensate for this.

Native TolC contains no cysteines with which to form higher molecular weight bands, so any additional higher molecular weight bands that are present in the membranes with TolC cysteine mutant membranes, but not native TolC membranes, must be due to the interaction between TamA and TolC cysteines. Intriguingly, all TolC cysteine mutants could interact with the last β-strand of TamA, but represented at least 3 different higher molecular weight bands (between approximately 110 and 150 kDa). These bands could be readily distinguished from non-TolC-containing bands (white triangles, Fig 3C), which likely represent TamA–TamA interactions, suggesting that TamA is inserting nascent TamA molecules into the outer membrane. The presence of multiple higher molecular weight bands that migrate at different speeds can be attributed to the physical location of the cysteine mutations in both TamA and TolC, which impacts gel mobility because they no longer form a linear molecule. For example, the TamA-G553C places the cysteine mutation at the far C-terminus of TamA, whereas TolC-G253C places the cysteine mutation in the middle of TolC. When a disulphide interaction forms between these 2 molecules, the relative structure could resemble a "T" shape (with TamA and TolC molecules arranged 90˚ relative to each other). Conversely, TolC-G45C places the cysteine mutation at the N-terminus of TolC, so instead of a "T" shape, an "L" shape could form. In both cases, if TamA and TolC molecules were arranged 180˚ relative to each other, the structure could resemble a more linear molecule, thus explaining the different electrophoretic mobilities observed.

To ensure that this interaction was physiologically relevant, we sought to distinguish between interactions mediated by TamA molecules integrally inserted into the outer membrane and nascent TamA molecules that have remained associated with the membrane during purification. To do this, we performed a Triton X-114 extraction of membrane proteins (see Materials and methods). Above its relatively low cloud point (20˚C), Triton X-114 solutions can be used to separate integral (BamA, Fig 3C) and peripherally associated (DnaK, Fig 3C) membrane proteins into detergent-enriched and aqueous phases, respectively. While all disulphide interactions were predominantly mediated by TamA molecules that were integrally inserted into the outer membrane, we found that the interaction with nascent TolC-S287C was so prominent that it even occurred with nascent TamA molecules that had not yet fully assembled themselves. To assess whether these interactions were correctly localised to the outer membrane, we performed sucrose density fractionation (S5A Fig). The inner membrane protein PpiD was enriched within fractions 1 to 3, whereas the outer membrane protein OmpF was enriched in fractions 5 to 7 (S5B Fig). Further analysis of fractions 1 and 7 revealed that the higher molecular weight bands of TamA-G553C/TolC-S287C were enriched in the outer membrane (lanes 4 and 8, S5C Fig).

We next sought to explore whether a stepwise insertion mechanism could be inferred by assessing how TolC interacts with the first β-strand of the lateral gate. To do this, we paired each TolC construct with TamA-G252C (i.e., cysteine substitution in the first transmembrane β-strand). In this scenario, we expected that after engaging the terminal TolC transmembrane β-strand, the first β-strand of TamA would be sterically blocked from interacting with the 3 other TolC strands; however, we found that this was not the case (Fig 3D). Instead, all 4 TolC cysteine mutants could interact with the first β-strand of TamA in at least 3 different higher molecular weight bands (between approximately 150 and 225 kDa) as assessed by nonreducing SDS-PAGE (black triangles, Fig 3D), suggesting that rather than a stepwise threading mechanism, TamA can initiate assembly from any strand, but with a clear preference towards its substrate's terminal β-strand given that the aqueous phase had an abundance of TamA-G252C/TolC-S287C species (Fig 3D). Additionally, the higher molecular weight bands formed by TamA-G252C/TolC-S287C were found to be enriched in the outer membrane fraction following sucrose density fractionation (lanes 2 and 6, S5C Fig). Together, these data reveal that the

lateral gate of TamA (which is integrally inserted into the outer membrane) can interact directly with TolC molecules.

## Phylogenetic analysis of TolC-like proteins

We next sought to assess the prevalence of TolC-like proteins among all bacteria. To do this, the duplicated OEP (outer membrane efflux protein) fold domain feature characteristic of TolC-like proteins (Fig 1C) was used as a probe and 170,554 TolC-like sequences were identified from across a range of bacterial species. This sequence dataset was refined to 518 exemplar TolC-like sequences using CD-HIT [43] (see Materials and methods; S1 Data) before further refinement using SignalP 5.0 [44] reduced this number to 416 secreted proteins. The SignalP step was used to ensure that the TolC-like proteins were secreted to distinguish them from the evolutionarily related periplasmic adaptor proteins [30], which are instead anchored to the inner (cytoplasmic) membrane. Finally, a phylogenetic tree was generated and revealed that TolC-like proteins exist among all gram-negative bacteria with a lipopolysaccharide (LPS)-containing outer membrane, as well as in the early-branching Thermotogae that have distinct LPS-free outer toga membranes (Fig 4A; S1 Data). There were small clusters of the Firmicutes and Proteobacteria (Fig 4A, green and white, respectively) within specific tree nodes indicative of vertical inheritance of those TolC-like proteins. However, all Phyla with more than one protein represented had proteins interspersed across various nodes within the tree, suggesting multiple acquisition events of these diverse TolC-like proteins consistent with HGT events (Fig 4A; S1 Data).

To delineate the spread of the archetypal sequence (orange arrow, Fig 4A), we further investigated the CD-HIT cluster that TolC represented. There were 6,667 proteins identified within this cluster, 6,642 of which were from Proteobacterial species. This cluster was reanalysed using stricter identity cutoffs (between 0.70 and 0.95), and TolC was only ever found clustered within species of the Enterobacterales Order within the Gammaproteobacteria. Either the TolC archetype has sequence or environmental features limiting its distribution, or it is a more recently evolved form in the TolC-like protein family. In either case, the tree makes clear that while TolC might be a heavily studied member of the family, it is a minor member of a highly prevalent protein family that has widespread sequence diversity.

Within the Firmicutes node, TolC-like proteins were not only present in the gram-negative species (like the Negativicutes) but were surprisingly widespread among species that only contain a single membrane (i.e., monoderms). On further examination, many of these proteins were found in operons with homologues of the other 2 tripartite pump components (S6A Fig). Due to the obvious absence of an outer membrane for the TolC-like protein to reside in, we thought that an alternative structure that resembles an outer membrane may house the TolC-like protein instead. In monoderm bacteria, we thought this would either be the endospore or the bacterial surface layer (S-layer), a crystalline protein coat that can play a role in adherence to host cells and protection against antibacterial agents [45]. In each of the examples, we noticed that the TolC-like protein was only present in bacteria that contained an S-layer (or genes encoding putative S-layer proteins), whereas the ability to form endospores was variable (Fig 4B).

Two previous studies have demonstrated the presence of TolC-like proteins in the secretomes of *Clostridioides* (formerly *Clostridium*) *difficile* [46,47] along with abundant amounts of S-layer proteins. During high toxin production, *C. difficile* sheds a lot of its S-layer content, along with its TolC-like protein, further suggesting that its TolC-like protein may be localised to the S-layer [46]. We next sought to use homology modelling to structurally characterise the TolC-like protein from *C. difficile* using Phyre2 [48]. Not surprisingly, it was modelled against

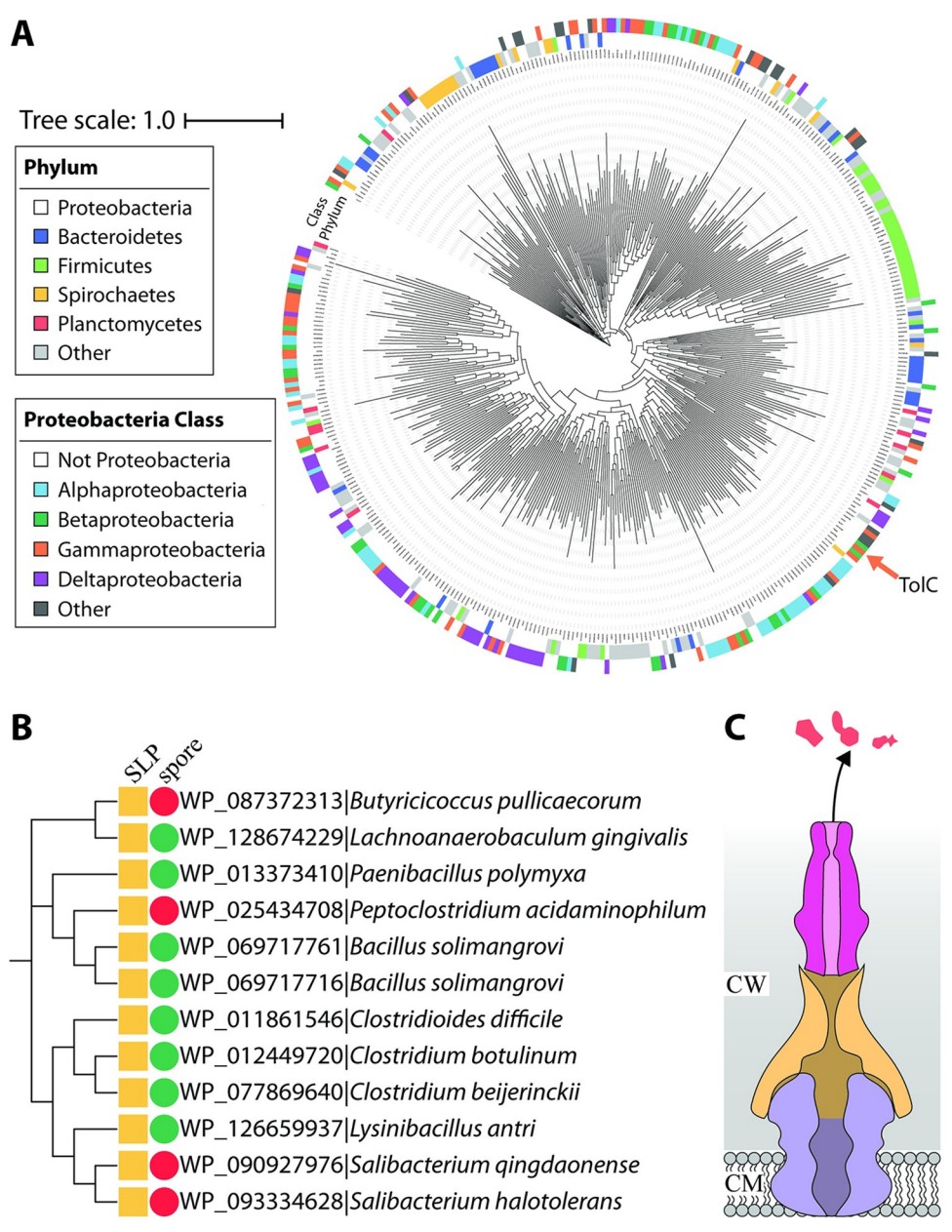

**Fig 4. Distribution of TolC-like proteins.** (A) Phylogenetic tree of 416 TolC-like proteins (S1 Data) from across the range of bacterial Phyla. The archetypal *E. coli* TolC (accession: P02930) is indicated with an orange arrow. (B) Cladogram of 12 TolC-like proteins found within 5 different families of monoderm Firmicutes. Protein ID is shown alongside the species' name. Presence of proven or putative SLPs is indicated by a yellow square. Endospore-forming ability (green circle) or inability (red circle) is also indicated. (C) Architecture of a TolC-dependent efflux pump in monoderm Firmicutes. CW, cell wall (i.e., peptidoglycan and S-layer); CM, cytoplasmic membrane; SLP, S-layer protein.

TolC (and other TolC-like proteins) but had a very intriguing omission in its β-barrel domain (S6B–S6E Fig). While the β-barrel domain in gram-negative bacteria sit within a hydrophobic outer membrane and so have a hydrophobic surface, the surface of the modelled β-barrel domain was still hydrophilic, further suggesting that it more likely sits within the proteinaceous S-layer (and not the bacterial endospore) (Fig 4C).

Considering that some bacteria must use their β-assembly machinery in the form of the BAM complex and the TAM to help assemble their TolC-like proteins, we wanted to assess whether TolC-like proteins are encoded together with homologues of the BAM complex and/or the TAM. To do this, we systematically searched through each bacterial Phyla to determine if proteins resembling BamA, TamA, or TamB could be encoded (Fig 5; S7 Fig). With the sole exception of the monodermic Firmicutes, all TolC-containing bacteria (31/37 Phyla) could encode a BamA homologue (31/37 Phyla) and most also encoded a TamB homologue (26/37 Phyla). In those bacterial Phyla, it is thought that TamB forms a complex with BamA instead [20,28,49]. Because TamA evolved more recently from a duplication of BamA [20,28], only 14/37 Phyla encoded a TamA homologue. Perhaps not surprisingly, the monodermic Firmicutes did not encode BAM, begging the question "How do the TolC-like proteins in these species assemble?"

## Discussion

In Proteobacteria, both the BAM complex and TAM work synergistically to assemble the vast majority of β-barrel proteins into the outer membrane [17]. Multiple BAM complexes coalesce into assembly precincts in order to assemble oligomeric proteins [24], and native PAGE analysis indicates that the TAM forms stable higher order complexes of both approximately 400 kDa (dimer of TAM) and approximately 600 kDa (trimer of TAM) [25,39]. Previously, we have shown that the TAM is especially important for the assembly of monomeric (e.g., Ag43) and dimeric (e.g., FimD, intimin) proteins [25,38,51] and now extend this to include trimeric TolC-like proteins.

The cross-linking assay used here included an additional control consisting of native TolC to distinguish TamA–TamA disulphide interactions (white arrows, Fig 3C and 3D). While this could be an example of TamA inserting TamA substrate molecules into the outer membrane, it could alternatively be an example of an interaction between the stable higher order structures of TAM dimers and trimers. In one model for TolC assembly, a trimer of TAMs would each handle one of the 3 protomers assembled into the TolC trimer, where each TAM would be responsible for a single TolC stave domain. The TolC stave domains would then need to stitch together at the interface between the fourth β-strand of one stave domain and the first β-strand of an incoming stave domain. Consistent with such a step in the model, the lateral gate had a stronger preference for the fourth β-strand of TolC (to assist the threading of an incoming stave domain) and the relatively weaker preference for the first β-strand of TolC (which can be more easily released and bonded to a neighbouring stave domain).

The current model for BamA-mediated threading of substrate molecules into the outer membrane is exemplified by a hybrid structure of a stalled BAM complex assembling a defective BamA substrate [52]. In this hybrid structure, the terminal β-strand of the lateral gate does not pair with the incoming β-strands as previously thought [53] but is instead folded inwardly to promote eventual substrate budding from the lateral gate into the outer membrane [52]. However, despite the structural similarities between BamA and TamA [17], our data with TolC are not consistent with this newly proposed mechanism of substrate insertion. Instead, we show that all 4 β-strands of the TolC stave domain can interact with both strands of the TamA lateral gate. We cannot be certain whether this promiscuity is an inherent function of TolC needing to form stable trimers prior to insertion into the outer membrane or of the TAM, which has been shown to be especially promiscuous in its ability to insert foreign substrates [16]. In either case, this promiscuity is especially amenable for ensuring that efflux pumps that may be acquired by HGT can be immediately assembled without the need for adaptation or alternative dedicated machinery.

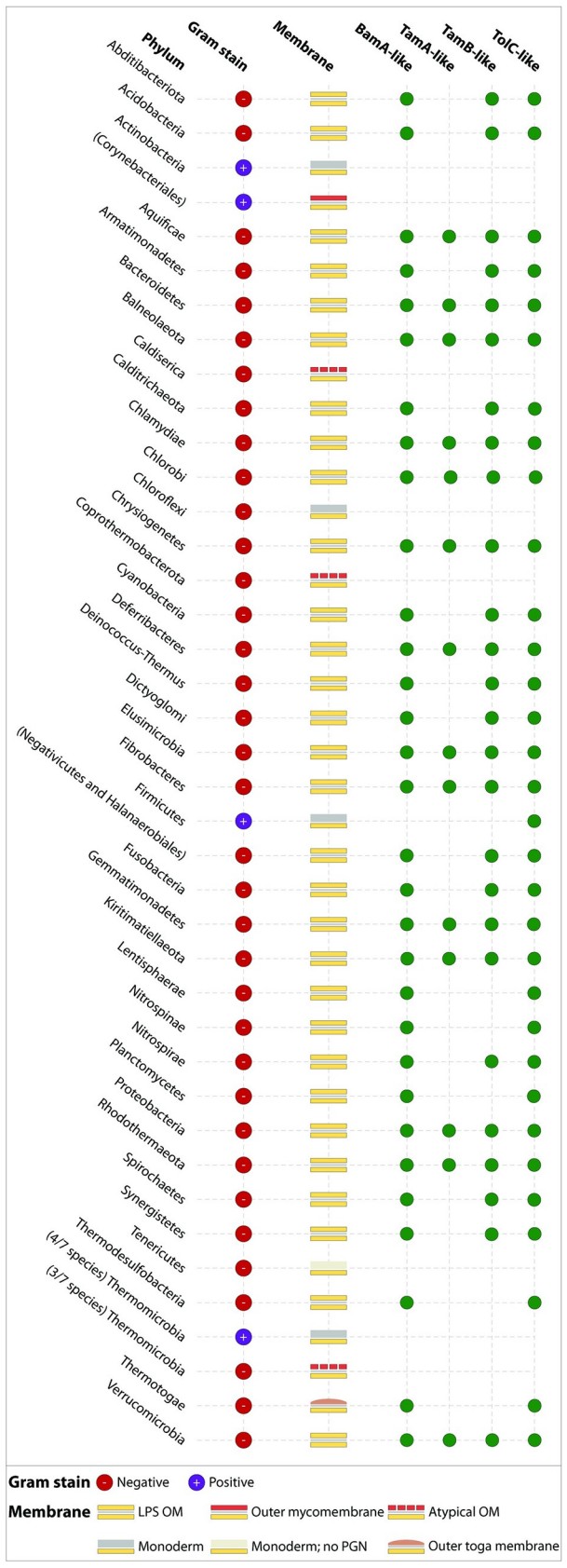

**Fig 5. Distribution of TolC and β-barrel assembly factors across all bacterial Phyla.** Using the list of 37 bacterial Phyla catalogued on LPSN as at 26 Feb 2021 [50], we systematically determined whether BamA, TamA, TamB, and TolC homologues were encoded within each Phyla (green circles) or are absent (empty). Data underlying this figure are either (i) presented in Fig 4A and S1 Data (for TolC); (ii) presented in S7 Fig (including accession IDs where relevant); or (iii) has otherwise been published previously [20,28]. Due to the unusual characteristics of the Actinobacteria and Firmicutes, we have separately listed the Orders and Class with divergent membrane profiles in parentheses below their respective Phyla. For the Thermomicrobia, we have separately listed the number of validly published species with the indicated membrane profiles. LPS, lipopolysaccharide; OM, outer membrane; PGN, peptidoglycan.

Resistance to antibiotics is spread by HGT, and gene cassettes that encode efflux pumps are recognised as being distributed by these horizontal gene flows. However, enacting a drug-resistant phenotype depends on the "alien" protein sequences delivered through HGT being assembled into a correct inner membrane–and outer membrane–spanning architecture. The phylogeny we constructed shows that discrete, nonidentical clusters of TolC-like protein sequences have been transferred across species (horizontally) and also inherited over evolutionary time (vertically) within Phyla. Evaluating the presence of the membrane chaperones BamA, TamA, and TamB in these various Phyla that both the subunits of the TAM are widespread (Fig 5). The prevalence of the various TolC-like proteins with an αβ-barrel architecture is similarly widespread, correlating strongly with the presence of the efflux pumps with the chaperone required for their assembly. In the case of fimbrial usher proteins, which are also shared via HGT, the prospect that an alien usher protein can only be established in a host that has a suitable TAM has been verified experimentally [16]. We suggest that this need for flexibility in chaperones is likewise true for TolC-like proteins. As such, future efforts to target efflux pump expression as a means to thwart AMR may also need to take into account the ability of both the BAM complex and the TAM to assemble TolC and the flexibility this affords bacteria in their quest to evade antibiotic exposure.

## Materials and methods

### Strains, plasmids, primers, and growth conditions

Strains, plasmids, and primers used in this study are listed in S2–S4 Tables, respectively. *E. coli* strains were routinely incubated in lysogeny broth (1% w/v tryptone, 0.5% w/v yeast extract, 0.5% w/v NaCl), and unless otherwise indicated, incubation was performed at 37°C at 200 strokes per min (25 mm orbit). When solid media was required, 1.5% w/v bacteriological agar was added and growth was 37°C. Media was supplemented with 100 μg mL$^{-1}$ ampicillin and/ or 34 μg mL$^{-1}$ chloramphenicol for plasmid maintenance or 30 μg mL$^{-1}$ kanamycin for strain maintenance (only for strains without plasmids). *E. coli* BL21 Star (DE3) Δ*tolC*::*kan* was generated using the linear fragment method [54] with a linear PCR product (CSP418/CSP419; 1,749 bp product) comprising the kanamycin resistance cassette and surrounding homologous regions from BW25113 Δ*tolC*::*kan* [55] and subsequently confirmed by PCR (CSP420/ CSP188; 736 bp product). Plasmids were generated as indicated in S3 Table. Amino acid substitutions are numbered according to the mature protein sequence where the first residue is the +1 residue following signal peptide cleavage.

### Antibody production

All antibodies and antibody concentrations used in this study are listed in S5 Table. *E. coli* C41 (DE3) cells were transformed with either pCJS106 (*surA* expression), pCJS158 (*tolC* expression), or pPpiD (*ppiD* expression) (see S3 Table). Cells were grown in 1 L of Terrific Broth (12 g/L tryptone, 24 g/L yeast extract, 0.4% v/v glycerol, 2.31 g/L KH$_2$PO$_4$, and 12.84 g/L K$_2$HPO$_4$)

with shaking until cultures reached an optical density ($OD_{600}$) of 0.8, at which point the temperature was lowered to 18°C and protein expression was induced with 0.2 mM IPTG overnight. Cells were collected by centrifugation, pellets resuspended in lysis buffer (50 mM Tris (pH 8.0), 400 mM NaCl, 2 mM $MgCl_2$, and 20 mM imidazole) and cells lysed via 2 passes through an Avestin Emulsiflex C3 cell press. In the case of the soluble periplasmic proteins SurA and PpiD (i.e., no inner membrane anchor), the postcentrifugation lysate was used for further Ni-affinity and size exclusion purification. In the case of the membrane-associated TolC, the postcentrifugation lysate was subjected to a further high-speed centrifugation (Ti-45 rotor, 38,000 rpm, 60 min, 4°C) to isolate membranes containing the recombinant protein. Membranes were resuspended in 50 mM Tris (pH 8.0), 400 mM NaCl, 10% Elugent, and the soluble membrane fraction applied to the Ni-affinity column as follows (with the addition of 0.03% DDM in all subsequent steps). His-tagged proteins were first purified by application of soluble lysate fraction over Ni-affinity chromatography, with lysis buffer used for binding to the 5-mL nickel HisTrap HP column (GE Healthcare) and, following washing, elution using a gradient of elution buffer (50 mM Tris (pH 8.0), 400 mM NaCl, and 1 M imidazole) to elute each protein, respectively. Proteins were further purified by size exclusion chromatography using a HiLoad 16/600 Superdex 200 pg column (GE Healthcare) equilibrated in 25 mM Tris (pH 8.0) and 200 mM NaCl. Protein presence in each peak was assessed by Coomassie-stained reducing SDS-PAGE and activity assays. SurA, PpiD, and TolC rabbit polyclonal antibodies were generated to the respective purified recombinant protein at the Monash Animal Research Platform (MARP) in adherence to their ethics approved protocols for generating antibodies in rabbits (this work is specifically covered by ERM project # 14152).

## SDS-PAGE and immunoblotting

SDS-PAGE (tris-glycine buffer system with 8% gels) and subsequent immunoblotting was performed using BioRad or BioCraft equipment (according to the manufacturer's recommendations), with the following exceptions. Nonreducing SDS loading buffer comprised 50 mM Tris–HCl (pH 6.8), 1% w/v SDS, 10% v/v glycerol, and 0.005% w/v bromophenol blue. Reducing SDS loading buffer additionally contained 100 mM DTT. After analysis by nonreducing SDS-PAGE, protein gels were incubated with in-gel denaturation buffer (4% w/v SDS, 50 mM DTT, 50 mM Tris–HCl [pH 6.8]). The in-gel denaturation buffer was preheated to 65°C and then immediately added to the protein gel, where it was incubated (and allowed to cool) for 20 min (37°C, 50 rpm in 25 mm orbit), before proteins were transferred to a 0.45-μm PVDF membrane.

## Pulse chase analysis

*E. coli* BL21 Star (DE3) wild-type or Δ*tamA*::*kan* strains with both a *fimD* expression vector (pKS02) and either pACYCDuet-1 or one of 7 *tamA* expression vectors (see S3 Table) were subjected to pulse chase analysis using a [$^{35}$S]-radiolabelling technique described previously [38], with a few modifications. Pulse labelling was for 45 s using 0.74 MBq mL$^{-1}$ of EXPRE$^{35}$S$^{35}$ Protein Labeling Mix (NEG072, Perkin Elmer), and chase media was supplemented with 0.5 mg mL$^{-1}$ rifampicin. For pulse chase analyses using reducing or oxidising conditions, minimal media were also supplemented with 5 mM DTT or 100 μM $CuSO_4$, respectively. Radiation was captured for up to 20 h using a storage phosphor screen (GE Healthcare) and analysed using an Amersham Typhoon 5 Biomolecular Imager. Each pulse chase analysis was performed with at least 3 biological replicates (i.e., from 3 independent batches of pulse chase ready samples).

## Densitometry

To approximate the amount of oxidised (locked lateral gate) and reduced (open lateral gate) TamA, samples ready for pulse chase analysis (i.e., those that had been snap frozen in liquid nitrogen after sulphur starvation [38]) were thawed on ice and subjected to centrifugation (3,000 ×$g$, 5 min, 4°C). Cell pellets were resuspended in 50 μL of nonreducing SDS loading buffer and analysed by SDS-PAGE using an alternative acrylamide/bis ratio to resolve the oxidised and reduced forms of TamA: 2.6% cross-linker (instead of 3.3%). Immunoblotting was performed as described above. Densitometry was performed using ImageJ 1.51r [56] according to manufacturer's instructions, and the proportion of reduced and oxidised forms was averaged. Densitometrical analysis was on immunoblots prepared using 3 independent batches of pulse chase ready samples (S1 Fig).

## MIC determination

*E. coli* BL21 Star (DE3) wild-type or Δ*tolC*::*kan* strains were transformed with pET-15b or one of 5 *tolC* expression vectors (see S3 Table), and each transformant was considered to be a biological replicate for the purposes of MIC determination. MICs were taken as the lowest concentration of erythromycin where there was no visible bacterial growth and were performed using the microdilution method described previously [57], with some modifications. The dilution range of erythromycin was 0.25 μg mL$^{-1}$ to 128 μg mL$^{-1}$, the initial inoculum of *E. coli* was normalised to an optical density at 600 nm of 0.0005, and LB media supplemented with 0.02 mM IPTG was used. MICs were performed in biological triplicate.

## Membrane isolation and sucrose density fractionation

*E. coli* BL21 Star (DE3) Δ*tamA*::*kan* with one of 5 *tolC* expression vectors (see S3 Table) and one of 2 *tamA* expression vectors (pCJS129 containing TamA-G252C or pCJS134 containing TamA-G553C) were diluted 1:50 from saturated overnight cultures into 200 mL media and incubated to early log phase (OD$_{600}$ of approximately 0.30). Expression was induced on addition of 0.2 mM IPTG, and cells were incubated for a further 1 to 2 h before harvesting (5,000 ×$g$, 10 min, 4°C). Membranes were subsequently purified by ultracentrifugation as previously described [58], and protein concentrations were quantified by NanoDrop. Three independent batches of membranes were prepared for analysis, and their purity was confirmed by immunoblot (S3 Fig). Sucrose density fractionation was performed as previously described [58], using OmpF and PpiD antibodies to demonstrate the relative positions of the outer membrane and inner membrane, respectively.

## Triton X-114 extraction

Precondensation of Triton X-114 was performed to remove impurities as described previously [59]. Membranes were thawed on ice, and 2 aliquots (7.2 μg and 250 μg) were transferred to fresh tubes. The smaller volume was diluted to 20 μL with water and added to 20 μL of 2× SDS loading buffer. These samples represent "Total" membranes. Following Triton X-114 extraction [59] of the remaining 250 μg membrane samples, 2× SDS loading buffer was used to dilute detergent-enriched and aqueous phases at a ratio of 20:1 and 1:1, respectively (buffer:sample). Samples were then analysed by SDS-PAGE and immunoblotting (as described above) with antibodies raised against TamA (nonreducing SDS-PAGE) and TolC (nonreducing SDS-PAGE) or BamA (reducing SDS-PAGE; integral membrane protein control) and DnaK (reducing SDS-PAGE; peripheral membrane protein control). Antibodies raised against F$_1$β (another peripheral membrane protein control) were used in place of DnaK for some of the

replicates. This experiment was performed in biological triplicate, from 3 independent batches of membrane samples.

## Phylogenetic tree

All protein sequences containing the OEP fold characteristic of TolC-like proteins (Pfam: PF02321) were obtained from InterPro (version 84.0) [60]. A total of 170,554 sequences were obtained on February 16, 2021, and their associated taxonomy information were obtained from the UniProt database [61,62]. Proteins with incomplete taxonomy information and those with Candidatus status were removed, leaving 88,248 proteins with complete taxonomy. CD-HIT [43] was subsequently employed to remove redundant sequences using an identity cutoff of 0.1, and after manual inspection of the remaining 520 proteins, a further two were removed. TolC-like proteins are thought to have evolved from periplasmic adaptor proteins [30], so SignalP 5.0 [44] was used to distinguish TolC-like proteins from potential periplasmic adaptor protein contaminants. Proteins from Firmicutes were filtered using the gram-positive protocol, while all others used the gram-negative protocol. Of 88 predicted lipoproteins, three (protein ID: M3JEC1, A0A5C1QEM5, A0A1H3BRV2) did not have the N-terminal cysteine in the mature protein sequence required for acylation and were instead considered to be standard proteins with a signal peptide. Proteins that were predicted to have a signal peptide ($n = 331$) or a lipoprotein domain without the inner membrane retention signal ($n = 85$) were designated as TolC-like proteins. Proteins that were not predicted to have a signal peptide ($n = 118$) or predicted to have a lipoprotein domain with an inner membrane retention signal ($n = 0$) were removed because they likely resembled periplasmic adaptor proteins that are anchored to the inner (cytoplasmic) membrane. A multiple sequence alignment of the remaining 416 proteins (S1 Data) was generated using MAFFT (version 7.402) [63] using the default settings, except 2 parameters (-localpair—maxiterate 1,000). A phylogenetic tree was inferred using FastTree (version 2.1.10) [64]. For the 12 TolC-like proteins found within monodermic Firmicutes, a multiple sequence alignment (Clustal Omega [65], default settings) and cladogram were generated using the EMBL-EBI online resource [66]. All trees were visualised using iTOL (version 6.4) [67].

## Supporting information

**S1 Data. Full details of each leaf from the phylogenetic tree shown in Fig 4A.** UniProt accession numbers appear in the order they do in the phylogenetic tree (Fig 4A).
(XLSX)

**S2 Data. Densitometry data underlying S1B Fig.**
(XLSX)

**S1 Fig. Oxidation states of TamA cysteine mutants.** (A) Cells prepared for pulse chase analysis were subjected to nonreducing SDS-PAGE and immunoblotting using antibodies raised against TamA. TamA oxidation states are indicated, based on their migration rates. Biological replicate number is indicated to the left of immunoblots. Uncropped images are presented in S1 Raw Images; original immunoblots are presented in S3 Raw Images. (B) Densitometry of "lockable" TamA was determined using ImageJ 1.51r. Values displayed are equal to the density for oxidised species as a percentage of both oxidised and reduced species. Each biological replicate ($n = 3$) is shown, with mean indicated by dashed lines and error bars representing standard deviation. Data underlying this figure are presented in S2 Data.
(TIF)

**S2 Fig. FimD assembly as a readout of TamA function.** (A) Schematic of the pulse chase experiment based on [38]. *E. coli* BL21 Star (DE3) cells harbouring a plasmid with the gene of interest (orange arrow) are subjected to sulphur starvation to deplete sulphur-containing amino acids. Cells are then subjected to rifampicin treatment (1 h) to block native RNA transcription before IPTG induction (5 min) allows transcription from the T7 RNA polymerase (which is not sensitive to rifampicin) promoter upstream from the plasmid-encoded gene of interest. Cells are pulsed (45 s) with [$^{35}$S]-methionine and [$^{35}$S]-cysteine before chase media (containing [$^{32}$S]-methionine and [$^{32}$S]-cysteine) is added. On addition of extracellular protease (PK, yellow), protease-sensitive radiolabelled proteins can be detected once they are localised to the outer membrane through the accumulation of degradation products or a reduction in full-length protein. (B) FimD assembly was monitored over time by pulse chase analysis in the indicated strains of *E. coli* BL21 Star (DE3) harbouring pKS02 (*fimD* expression vector) and either pACYCDuet-1 (base vector) or pCJS69 (*tamA* complementation vector). (C) FimD assembly was monitored as per panel B, except all strains were *E. coli* BL21 Star (DE3) Δ*tamA* harbouring pKS02 and the plasmid encoding the indicated TamA cysteine mutants. Media were supplemented with the indicated reducing or oxidising agent, or they were not supplemented (i.e., nonreducing) as indicated. (B-C) Aliquots were taken at 10 s (0 min), 2, 4, 8, 16, and 32 min (panel B) or 8 min only (panel C) and treated with (+) or without (−) proteinase K. Total protein was analysed by SDS-PAGE and storage phosphor imaging. The position of FimD, TamA, and its fragments A, B, and C are indicated to the right of the autoradiograms, and protein standards are indicated on the left (sizes are in kDa). The presence of native TamA is indicated as a cartoon to the left (panel B only). (B-C) Uncropped images are presented in S1 Raw Images; original autoradiographs (including relevant replicates) are presented in S2 Raw Images.
(TIF)

**S3 Fig. Purity of membranes.** During membrane isolation [58], 1 mL aliquots were taken immediately before ultracentrifugation (Total = "T") and 1 mL aliquots of the supernatant were taken after ultracentrifugation (Supernatant = "S"). These aliquots were subjected to TCA precipitation and washed with acetone before resuspension in 100 μL SDS loading buffer. Membranes ("M") were diluted in SDS loading buffer so that the final protein concentration was 1 μg/μL. Samples were analysed by 10%, 12%, or 16% SDS-PAGE and immunoblotting to determine purity of membranes. The membrane protein control (αBamA) was found only in the total and membrane lanes, whereas the periplasmic control (αSurA) and cytoplasmic control (αGroES) were found only in the total and supernatant lanes. Uncropped images are presented in S1 Raw Images; original immunoblots are presented in S3 Raw Images.
(TIF)

**S4 Fig. TolC-TolC bands obfuscate TamA-TolC bands.** Total membranes were analysed by 8% nonreducing SDS-PAGE and immunoblotting using the indicated antibodies. Uncropped images are presented in S1 Raw Images; original immunoblots are presented in S3 Raw Images.
(TIF)

**S5 Fig. Sucrose density fractionation.** (A) Schematic of sucrose density fractionation. Membranes were isolated and subjected to a 6-step sucrose gradient as indicated (60%–35% w/w sucrose) by ultracentrifugation (200,000 ×*g*, 17 h, 4˚C). Twelve 1 mL fractions were then obtained using 70% w/w sucrose as the displacing fluid as indicated. (B) Fractions were analysed by 10% SDS-PAGE and immunoblotting for the inner membrane (αPpiD) or the outer membrane (αOmpF). (C) Fractions 1 and 7 were subjected to nonreducing SDS-PAGE as per

Fig 3C and 3D. Black triangles correspond to TamA–TolC interactions, whereas white triangles correspond to TamA–TamA interactions, as per Fig 3C and 3D. (B-C) Uncropped images are presented in S1 Raw Images; original immunoblots are presented in S3 Raw Images.
(TIF)

**S6 Fig. Identification of tripartite efflux pumps in monodermic Firmicutes.** (A) The genetic organisation of the 12 selected efflux pumps depicted in Fig 4B are shown (not to scale). The NCBI accession number is shown below the indicated bacterial strains, and the locus tags for each gene are also shown. Green (regulatory protein), pink (TolC-like protein), yellow (periplasmic adaptor protein), purple (cytoplasmic membrane channel component), white (hypothetical protein). The *L. gingivalis* operon encodes 2 putative TolC-like proteins; the most downstream is depicted in Fig 4B. (B) Ribbon diagram of trimeric TolC (PDB: 1EK9) from *E. coli*. One monomer is coloured pink; the other 2 are coloured white. (C) Surface structure of the pink monomer shown in panel B. The transmembrane region of the monomer is coloured using the YRB [68] scale that colours side-chain nitrogens (from R or K residues) red, side-chain oxygens (from D or E residues) blue, and carbon atoms likely to form hydrophobic interactions yellow. The remaining surface structure is coloured grey. (D-E) Phyre2 [48] was used to solve the homology model structure of the TolC-like protein from *C. difficile* 630. Surface structure of the *C. difficile* protein modelled against *E. coli* TolC (PDB: 1EK9) is coloured as in C, with (E) or without (D) a superimposed ribbon diagram of itself.
(TIF)

**S7 Fig. Identification of β-barrel assembly factors from new bacterial Phyla.** Data underlying Fig 5. The distribution of *tamA* (blue), *tamB* (green), and *bamA* (orange) homologues not previously reported by us for *bamA* and *tamA* homologues [20] or for *tamB* homologues [28]. For comparison, the archetypes from *E. coli* MG1655 are shown. NCBI accession IDs (and version numbers) are shown alongside strain names. Gene locus tags are displayed beneath the relevant gene of interest. Genes are coloured according to predicted function: red (chaperone); purple (LPS and/or phospholipid synthesis); white (sigma factor); grey (other). It should be noted that, while we previously reported the presence of TamA in the Fibrobacteres Phylum [20], the *C. alkaliphilus* sequence that has since been published shows a curious case of pseudogenisation of POTRA domains (dark orange) upstream from a putative *tamA* gene due to a premature stop codon.
(TIF)

**S1 Table. TolC mutants are still functional.** MICs of erythromycin for the indicated strain in the presence (0.02 mM) or absence (0 mM) of IPTG. MIC were performed in biological triplicate, where MIC results were identical between each replicate. MIC, minimum inhibitory concentration.
(XLSX)

**S2 Table. *E. coli* strains used in this study.**
(XLSX)

**S3 Table. Plasmids used in this study.** Base vector refers to unmodified commercial vector. Vectors were confirmed by sequencing using the following oligonucleotide primers (see S4 Table): T7promoter and T7terminator (all vectors synthesised in this study) and either CSP359 (TamA vectors synthesised in this study) or CSP368 (TolC vectors synthesised in this study).
(XLSX)

**S4 Table. Oligonucleotide primers used in this study.** The red nucleotides within the oligonucleotide primer sequence indicate the changes made that will cause a cysteine substitution. (XLSX)

**S5 Table. Antibodies used in this study.**
(XLSX)

**S1 Raw Images. Raw images of figure files.** The uncropped image files of the cropped autoradiographs and immunoblots displayed in the figures and supporting information figures in order of appearance. Apparent sizes in kDa are indicated on the left where applicable. A red box is used to indicate the cropped portion of the image that was displayed in the indicated figure or supporting information figure, which was usually resized to fit within the broader context of each panel.
(PDF)

**S2 Raw Images. Raw images of radiograph replicates.** FimD assembly was monitored over time by pulse chase analysis using *E. coli* BL21 Star (DE3) Δ*tamA* cells complemented with the indicated TamA cysteine mutants (on a pACYCDuet-1 vector backbone) and also harbouring pKS02 (the *fimD* expression vector). In total, 28 gels are depicted and annotated based on the complementing TamA cysteine mutant indicated at the top of the page. For gels #1, #5, #8, #11, #13, and #19, aliquots were taken at 10 s, 20 s, and every 20 s thereafter for a total of 180 s and treated with (+) or without (−) proteinase K (PK) as indicated. For gels #23–28, aliquots were taken at 8 min and treated with protease as indicated. Otherwise (for the remaining gels), aliquots were taken at 10 s (0 min), 2, 4, 8, 16, and 32 min and treated with (+) or without (−) proteinase K (PK) as indicated. Total protein was analysed by 8% SDS-PAGE (for gels #4, #10, #16–18, #20–22) or 12% SDS-PAGE (for the remaining gels) and storage phosphor imaging, where radiation was captured using Amersham Typhoon 5 Biomolecular Imager. Due to the nature of the imager, the "raw" gel file pixel intensity is relative to the most saturated peak scanned across the entire image, and in most cases, the raw gel appears very faint (and cannot be annotated by itself). Therefore, an "adjusted" gel file that could be annotated (i.e., level adjustment) appears alongside the "raw" gel file. Gels represent biological replicates, except gels #16 and #17, which include technical replicates for G250C/E555C nonreducing conditions. Gels that have been used in the main or supporting information figures are indicated with red text.
(PDF)

**S3 Raw Images. Raw images of membrane replicates.** Raw image files for all immunoblots. Protein standards markers (where shown) are indicated in kDa to the left of the immunoblots. In some cases, the cropped portion of the figure is shown using a red box and whether the image represents one of the main or supporting information figures is indicated in red text. Otherwise, the immunoblots represent biological replicates.
(PDF)

## Author Contributions

**Conceptualization:** Christopher J. Stubenrauch, Rebecca S. Bamert, Jiawei Wang.

**Data curation:** Christopher J. Stubenrauch, Jiawei Wang.

**Funding acquisition:** Trevor Lithgow.

**Investigation:** Christopher J. Stubenrauch, Rebecca S. Bamert, Jiawei Wang.

**Writing – original draft:** Christopher J. Stubenrauch, Rebecca S. Bamert, Trevor Lithgow.

**Writing – review & editing:** Christopher J. Stubenrauch, Trevor Lithgow.

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
