## [Editor Report · Decision Letter 0]

15 Apr 2021

Dear Dr. Stubenrauch, 

Thank you for submitting your manuscript entitled "Drug efflux pump assembly into bacterial outer membranes requires non-canonical chaperones." for consideration as a Research Article by PLOS Biology.

Your manuscript has now been evaluated by the PLOS Biology editorial staff, as well as by an academic editor with relevant expertise, and I am writing to let you know that we would like to send your submission out for external peer review.

Please re-submit your manuscript within two working days, i.e. by Apr 17 2021 11:59PM.

Kind regards,

Paula 

---

Associate Editor

PLOS Biology

---

## [Decision Letter · Decision Letter 1]

18 May 2021

Dear Dr. Stubenrauch,

Thank you very much for submitting your manuscript "Drug efflux pump assembly into bacterial outer membranes requires non-canonical chaperones." for consideration as a Research Article at PLOS Biology. Your manuscript has been evaluated by the PLOS Biology editors, an Academic Editor with relevant expertise, and by independent reviewers.

The reviews of your manuscript are appended below. You will see that the reviewers find the work potentially interesting. However, based on their specific comments and following discussion with the academic editor, I regret that we cannot accept the current version of the manuscript for publication. We remain interested in your study and we would be willing to consider resubmission of a comprehensively revised version that thoroughly addresses all the reviewers' comments. We cannot make any decision about publication until we have seen the revised manuscript and your response to the reviewers' comments. Your revised manuscript would be sent for further evaluation by the reviewers.

We consider that you need to conclusively show that TolC insertion into OM is catalysed by TamA in order for us to consider the manuscript forward. Specifically, performing complex immunoprecipitation (Co-IP) would be essential for the current narrative of the manuscript, for the direct demonstration of TolC – TamA interaction as well as of the physiological relevance of this interaction. Please, also address the other concerns from the reviewer, such as the issues from reviewers #1 and #2 with the localization of the proteins and how you distinguish inner membrane, periplasmic or outer membrane components. Also, reviewer #2 and #3 have some concerns and questions about the gels, and reviewer #3 has other concerns such as lack of controls. Please address all the reviewers concerns. 

We appreciate that these requests represent a great deal of extra work, and we are willing to relax our standard revision time to allow you six months to revise your manuscript. We expect to receive your revised manuscript within 6 months.

**IMPORTANT - SUBMITTING YOUR REVISION**

*Resubmission Checklist*

*Published Peer Review*

*PLOS Data Policy*

*Blot and Gel Data Policy*

Sincerely,

Paula

---

Paula Jauregui, PhD

Associate Editor

PLOS Biology

REVIEWS:

Reviewer #1: Structural Biology, Macromolecular Assemblies and Drug Efflux.

Reviewer #2: Membrane protein folding.

Reviewer #3: Structural biology and host-pathogen interactions.

Reviewer #1: TolC is key component require for efflux pumps, the relocation and success assemble of TolC the required for tripartite functioning in the bacteria. The author provide evidence for possible interactions between TolC and TAM complex, suggest TolC is one of the substrate for the complex and mutations effect on the TolC present on the outer membrane. Overall this paper is well written, provide an indirect evidence of TolC-TAM in vivo interactions. Certain experiments is required to further confirm author's conclusion in the paper. 

1) membrane separation result for overall, inner membrane and outer membrane all fractions in the bacteria with quantitive measurement for the target proteins. 

2)direct evidence between TAM and TolC use purified protein is required, in vitro CO-IP experiment will be strong evidence support for the conclusion in the research . 

Reviewer #2: This manuscript presents evidence for TolC insertion and folding by TamA. Evidence is provided for the pervasiveness of TolC homologs, for the use of a crosslinking assay to determine TamA insertion, for the interaction between TamA and TolC, and finally for the existence of BamA, TamA and TolC homologs in various phyla and classes. 

The manuscript is extremely thought provoking and clearly written. I would very much like to see it published here. While the paper is generally clear the discussion of TolC homologs could be further clarified. Moreover, the gels contain a lot of data some of which is not described in the manuscript.

The authors identify 170k homologs to TolC in a very wide variety of organisms from the pfam OEP fold. Previous work has identified homology between the outer membrane component and the periplasmic component of RND efflux pumps (PMID: 30057025) such that there are proteins that are close homologs of both OEPs and PEPs simultaneously. It is unclear to me if and how the pfam OEP fold distinguishes between the periplasmic and outer membrane components and if there is the possibility that PEPs are being inadvertently being considered OEPs.

In figure 2 D and E could the authors explain the two bands that periodically disappear and reappear between A and FimD.

In figure 2D and E the authors describe band B as a folding intermediate of FimD. If this were the case, I would have expected B to appear first before the A and C bands, but frequently B appears after A and C. Can the authors give me some intuition for why this would be the case?

In Figure 3C, why are there two forms of TolC-TamA for the cys mutants in the first 2 strands, but one form of TolC-TamA for the cys mutants in the latter two strands? Also, the authors describe the soluble TolC-TamA bond to the fourth strand of TolC as being attached to a nascent TamA. How can they exclude the possibility that it isn't a nascent TolC to a mature TamA or a nascent TamA to a nascent TolC?

In Figure 3D, in contrast to 3C there are two forms of TolC-TamA for the second two strands of TolC binding to TamA and only one form of TolC-TamA when the first two strands of TolC bind TamA. What would cause the switching between two and one form?

MIC data should be provided in the supplement if nowhere else.

Minor concerns:

The acronym OEP should be defined at its first use.

TolC channel in figure 1A appears somewhat yonic. This is, of course, an artistic decision, but the authors might consider making the inside of the channel the lighter pink and the outside of the channel the darker pink.

Reviewer #3: Studenrauch, et al. report an investigation of the role of the E. coli TamA, a homolog of BamA, in inserting the major outer membrane component of many bacterial efflux pumps, TolC, into the outer membrane. The study reveals the widespread distribution of TolC in bacteria with an LPS outer membrane, and reveals a combination of both vertical and horizontal evolution. The authors show that the evolution of TolC is somewhat coupled with the Tam and/or Bam systems. Using biochemical experiments, the authors reveal that TamA's lateral gate can be locked into a non-functional state, attenuating insertion of substrate FimD into the outer membrane. The authors then go on to show that TamA can interact with TolC at multiple different residues.

The title of this manuscript and abstract suggests that this work will show that TolC folding/insertion in the outer membrane is performed by TamA, which would be an important advance in the field. However, no evidence is presented indicating that TolC is at all dependent on TamA, so this claim is not supported by the data provided. They do, however, show that TamA modulates the insertion/conformation of FimD in the OM, and that TamA can interact with TolC when they are both over-expressed. While it is conceivable that TamA may indeed insert TolC, these conclusions are not currently well supported by the experimental data presented. Consequently, we think that the claims and framing of the manuscript require considerable reconfiguration (e.g., is this more a story about the lateral gate of TamA than about TolC insertion?), and is not ready for publication in its current form. Additional controls and alternative design of some experiments is required. The specific comments are as follows:

MAJOR COMMENTS:

ln 18-21: authors don't directly show that TolC insertion into OM is catalysed by TamA. As this is the primary claim made in the title and abstract, and it is unsupported, this is a fundamental problem that must be resolved (either substantially changing the claims and narrative, or by providing data to support those claims)

MINOR COMMENTS:

ln 67: mention that TolC is topologically different to other substrates of BAM complex - should expand on how they are different. Is this referring to the mix of poring and helical periplasmic domain? To the porin split into three parts?

ln 86: Could include more background on TamA - for example could expand and say there is already data that suggest it inserts proteins into the outer membrane? I don't think they actually ever say that TamA is a homolog of BamA.

Fig. 1C: what is an OEP domain? Should specify in figure legend and main text (not just in methods)

Fig. 1D: Label orange arrow something like "E. coli TolC" on figure?

ln 118-122: This is very interesting. If involved in translocating molecules across the S-layer, one might expect the "Omp" portion of TolC to be quite different since it would no longer be membrane spanning. Does anything jump out in this analysis? Alternatively, might many/all of the "monoderm" species encoding TolC form spores, and perhaps these TolC proteins insert into the spore OM? I think C. difficile at least forms double-membraned spores.

ln 124-177: I would find it helpful to have the overall experimental design for the pulse-chase laid out a bit more clearly (and/or a schematic in Fig. 2). The Cys locking aspect was clear, and the differential protease sensitivity was clear. But what is being pulsed and why, and the purpose of the chase aren't stated, and required some detective work to figure out. The chase doesn't really seem to be doing much? Also, text refers to bands of ~40, 45, and 50 kDa, but the markers don't really match this. Perhaps the uppermost band should be called ~47 kDa instead of 50? It seems like the pulse-chase is being down in BL21 while over-expressing? I think that should be made more clear in the main text, I think I had to look up past papers to figure out this in a more native context.

ln 138: should this be an "unlocked" conformation rather than "unlockable"? If it's truly unlockable, then why is that? Using "unlockable" for both the unlocked form of double cysteine mutants but also to refer to single cysteine mutants is confusing. Perhaps just use "locked" and "unlocked" instead. Alternatively, could force the protein to be fully locked by promoting disulfide bond formation. For example, see this reference for similar experiments done with BamA using CuSO4 and copper-o-phenanthroline (see Noinaj et al, Structure, 2014). 

ln 138: should be "S1 fig" rather than "S1 data"?

ln 166: I don't understand how FimD can still be cleaved into a 45 kDa sized fragment in the absence of TamA.

Fig. 2: Would be nice to have the WT control as a reference here in the main fig. 

Fig 2C and D: It looks like the different gels have different exposures. It would be better to have samples be compared on the same gel/phosphor screen. This is probably more important than having all the time points, as most of the time points look very similar. Band B is indeed enriched in the "locked" mutants, but I do not believe it is completely absent in some of the single mutants. This is ok because TamA could have a backlog of FimD that hasn't been inserted yet - but this should be acknowledged? Seeing all 3 biological replicates somewhere for these experiments would also help.

ln 171-172: authors state that in the double mutants that FimD is predominantly the 45 kDa band and there are small amounts of the 40/50. The 45kDa one is enriched, but there is still plenty of 40/50. 

lns 210-212 - not sure I agree that all mutants can interact - the gels are a bit messy and I only really find S287C to be the very clear one.

Figure 3C and 3D experiments: The gels do not have controls to confirm that band shifts are due to TolC interaction. If authors can obtain antibodies against TolC (or against its tag), this would make the result more believable. This would allow us to compare TolC levels in the input, which is currently missing. Furthermore, it is surprising to me that all of these mutants would form disulfides to some degree. Perhaps some controls are needed to make sure this reflects protein-protein interactions and not just co-over-expression in periplasm? Ln 225-228 might be consistent with this sort of problem.

Fig. 4: More "data" than discussion? Maybe this should be presented in the context of other bioinformatic analyses?

---

## [Decision Letter · Decision Letter 2]

16 Dec 2021

Dear Dr. Stubenrauch,

Thank you for submitting your revised Research Article entitled "A non-canonical chaperone interacts with drug efflux pumps during their assembly into bacterial outer membranes." for publication in PLOS Biology. I have now obtained advice from some of the original reviewers and have discussed their comments with the Academic Editor. 

Based on the reviews, we will probably accept this manuscript for publication, provided you satisfactorily address following data and other policy-related requests.

DATA POLICY:

Regardless of the method selected, please ensure that you provide the individual numerical values that underlie the summary data displayed in the following figure panels as they are essential for readers to assess your analysis and to reproduce it: Figures 5 and S1B.

**Please also ensure that figure legends in your manuscript include information on where the underlying data can be found, and ensure your supplemental data file/s has a legend.**

We expect to receive your revised manuscript within two weeks.

*Published Peer Review History*

*Early Version*

Sincerely,

Paula

---

Associate Editor,

pjaureguionieva@plos.org,

PLOS Biology

Reviewer remarks:

Reviewer #1: Author answers all questions, I am happy with revision.

---

## [Editor Report · Decision Letter 3]

22 Dec 2021

Dear Dr. Stubenrauch,

On behalf of my colleagues and the Academic Editor, Csaba Pál, I am pleased to say that we can in principle accept your Research Article "A non-canonical chaperone interacts with drug efflux pumps during their assembly into bacterial outer membranes." for publication in PLOS Biology, provided you address any remaining formatting and reporting issues. These will be detailed in an email that will follow this letter and that you will usually receive within 2-3 business days (allow more time given the holiday season, it is likely you will receive it on January), during which time no action is required from you. Please note that we will not be able to formally accept your manuscript and schedule it for publication until you have any requested changes.

PRESS

Sincerely, 

Paula

---

Paula Jauregui, PhD 

Associate Editor 

PLOS Biology
